

# Evaluation of soil moisture in CMIP5 simulations over contiguous United States using in situ and satellite observations

Shanshui Yuan[1], Steven M. Quiring[2]

[1]Climate Science Lab, Department of Geography, Texas A&M University, College Station, Texas 77843, United States
[2]Atmospheric Sciences Program, Department of Geography, The Ohio State University, Columbus, Ohio 43210, United States

*Correspondence to*: Shanshui Yuan (yuanshanshui@tamu.edu)

**Abstract.**

This study provides a comprehensive evaluation of soil moisture simulations in the Coupled Model Intercomparison Project
Phase 5 (CMIP5) extended historical experiment (2003 to 2012). Soil moisture from in situ and satellite sources are used to evaluate CMIP5 simulations in the contiguous United States (CONUS). Both near-surface (0–10 cm) and soil column (0–100 cm) simulations from more than 14 CMIP5 models are evaluated during the warm season (April–September). Multi-model ensemble means and the performance of individual models are assessed at a monthly time scale. Our results indicate that CMIP5 models can reproduce the seasonal variability in soil moisture over CONUS. However, the models tend to
overestimate the magnitude of both near-surface and soil-column soil moisture in the western U.S. and underestimate it in the eastern U.S. There are large variations in model performance, especially in the near-surface. There are significant regional and inter-model variations in performance. Results of a regional analysis show that in deeper soil layer, the CMIP5 soil moisture simulations tend to be most skillful in the southern U.S. Based on both the satellite-derived and in situ soil moisture, CESM1, CCSM4 and GFDL-ESM2M perform best in the 0–10 cm soil layer and CESM1, CCSM4, GFDL-
ESM2M and HadGEM2-ES perform best in the 0–100 cm soil layer.

**Keywords:** CMIP5 simulation, in situ soil moisture, remote sensed soil moisture, CONUS.



# 1 Introduction

Soil moisture plays a critical role in hydrological processes, land–atmosphere interactions and climate variability. Through controlling water mass transfer, soil moisture affects runoff (Penna et al., 2011;Latron and Gallart, 2008;Zhang et al., 2001) and evapotranspiration (Wetzel and Chang, 1987;Vivoni et al., 2008;Detto et al., 2006). Soil moisture also influences the

surface energy balance by affecting latent heat and ground fluxes (Ek and Holtslag, 2004;Ford and Quiring, 2014a). Soil moisture is one of the direct measures of drought used to assess future drought conditions in the latest IPCC report (Hartmann et al., 2013). Therefore, accurate soil moisture simulation is useful for many applications.

There are three main types of soil moisture data: in situ observations, remote sensing and model simulations. In situ observations provide point measurements at a variety of depths. The spatial and temporal coverage of in situ observations is

quite limited and each in situ network may utilize different instruments and calibration techniques. These factors make it more challenging to use in situ soil moisture, however recent developments have improved the utility of these measurements. For example, the International Soil Moisture Network (ISMN) (Dorigo et al., 2011), which was initiated in 2010, collects in situ soil moisture from more than 1400 station internationally and provides quality controlled hourly-to-weekly soil moisture data. The North American Soil Moisture Database (NASMD) (Quiring et al., 2016) provides quality controlled daily soil

moisture from approximately 1800 stations, most of which are located in the United States. NASMD has been used for validating the North American Land Data Assimilation System (NLDAS) (Xia et al., 2015b, a) and to examine the nature of land-atmosphere interactions (Ford et al., 2015b;Ford et al., 2015c;Wang et al., 2015). There are numerous other studies that use in situ soil moisture from NASMD and ISMN. Ford and Quiring (2014a) used quantile regression to examine the relationship between in situ soil moisture and extreme temperature in Oklahoma. They found the soil moisture anomalies can

be used for predicting the percent hot days in the following month. Ford et al. (2015a) found that soil moisture can also be used to predict the onset of flash drought events earlier in Oklahoma. Brocca et al. (2013) found in situ soil moisture can be used to improve daily precipitation estimation at the catchment scale.

Soil moisture observations from satellites remote sensing, such as the Soil Moisture and Ocean Salinity (SMOS) mission (Kerr et al., 2001), NASA's Aquarius (Le Vine et al., 2007) and Soil Moisture Active-Passive (SMAP) missions (Brown et

al., 2013) can provide global soil moisture data. Previous studies have shown that satellite-derived soil moisture can accurately capture the annual cycle (Albergel et al., 2012b;Brocca et al., 2011), however, the accuracy of the satellite-derived soil moisture varies significantly both geographically and from product to product (Fang et al., 2016;Wanders et al., 2012). Rötzer et al. (2015) investigated the spatial and temporal behavior of the SMOS and the MetOp-A Advanced Scatterometer (ASCAT) soil moisture. They demonstrated that SMOS is more strongly affected by temporally invariant factors, such as

topography and soil properties, while ASCAT soil moisture is influenced by temporally variant factors, such as precipitation and evaporation. To overcome the limitations of satellite-derived soil moisture estimates, assimilated satellite products have been developed. Renzullo et al. (2014) used the ensemble Kalman filter method to assimilate AMSR-E and ASCAT-derived soil moisture. They found that data assimilation can significantly improve the accuracy of root-zone soil moisture estimates.





A merged soil moisture product from active and passive sensors was released by the European Space Agency (ESA) in 2010 (Liu et al., 2011). This is a part of the program on the Global Monitoring of Essential Climate Variables (ECVs), and hereafter it will be referred to as ECV soil moisture. ECV soil moisture has been validated globally (Dorigo et al., 2015) and in regional studies in places such as in China (An et al., 2016) and East Africa (McNally et al., 2016). One of the primary
limitations of satellite-based approaches is that they can typically only measure water in the top few centimeters of the soil (Crow et al., 2012).

Model simulation from offline land surface models (Koster et al., 2009) and fully coupled general circulation models (GCMs) (Srinivasan et al., 2000) is another source of spatially continuous soil moisture at variety of depths. However, validation studies have shown that these models can have significant bias. Guo and Dirmeyer (2006) compared 11 land surface models
from the Second Global Soil Wetness Project (GSWP-2) and found that although models can reproduce soil moisture anomalies, they do not accurately simulate the absolute soil water content. (Xia et al., 2015b) evaluated four land surface models within the North-American Land Data Assimilation System Project Phase 2 (NLDAS-2). They concluded that Noah and VIC model are wetter while Mosaic and SAC are drier. Compared with land surface models, coupled GCMs are more commonly used to investigate soil moisture–atmosphere interactions (Seneviratne et al., 2010). Koster et al. (2004) is a
benchmark study of soil moisture–temperature and soil moisture–precipitation coupling strength using 12 GCMs in the Global Land–Atmosphere Coupling Experiment (GLACE). They identified three global "hot spots" where one finds strong land–atmosphere coupling. However, they also demonstrated that there are substantial inconsistencies in coupling strength between models. van den Hurk et al. (2010) used realistic soil moisture initializations in the second phase of GLACE (GLACE-2) to improve the forecast skill of summertime temperature and precipitation in Europe.

In 2012, the fifth phase of the Coupled Model Intercomparison Project (CMIP5) was completed to provide a state-of-the-art multi-model dataset for advancing the knowledge of climate variability and climate change (Taylor et al., 2012). Li et al. (2007) concluded, based on previous versions of the CMIP models, that these models have difficulty accurately simulating the seasonal cycle of soil moisture. They also found that improved simulation of solar radiation and precipitation leads to more accurate soil moisture simulations. Although the CMIP5 models have been used to investigate land–atmosphere
interactions (Dirmeyer et al., 2013;Seneviratne et al., 2013;May et al., 2015;Lorenz et al., 2016), to date, there has not been a comprehensive evaluation of the accuracy of the CMIP5 soil moisture simulations in the United States. Therefore, this paper will address this knowledge gap.

In this study, we evaluate CMIP5 soil moisture simulations in two soil layers (0 to 10 cm and 0 to 100 cm) over CONUS using in situ and satellite-derived soil moisture. We evaluate both individual models and the multi-model ensemble mean
using in situ soil moisture from 363 sites as well as satellite observations. A description of the data and methods used in this study are presented in section 2. This is followed by presentation of the results and discussion in section 3 and the limitations and conclusions are summarized in section 4 and section 5, respectively.



## 2  Data and Methods

### 2.1  Study Regions

We evaluate the CMIP5 soil moisture simulations over CONUS and in eight regions (Figure 1). These regions were defined using a land cover classification from U.S. Geological Survey (Loveland et al., 2000). These regions (dashed boxes in Figure 1) were utilized by Notaro et al. (2006) and they have been applied in other land–atmosphere studies (Mei and Wang, 2012;Sanchez-Mejia et al., 2014;Wu and Zhang, 2013). In this study, we made some small adjustments to these regions so that they included more in situ sites (solid boxes in Figure 1). The eight regions are: Midwest (MW: 38°–47.5° N, 94°–80° W), Northeast (NE: 38°–47.5° N, 80°–67° W), Northern Great Plains (NGP: 34.4°–49° N, 105°–94° W), Northern Shrubland (NS: 40°–49° N, 119.4°–105° W), Northwest (NW: 40°–49° N, 124°–119.4° W), Southeast (SE: 30°–38° N, 92.5°–75° W), Southern Great Plains (SGP: 25°–34.4° N, 105°–94° W) and Southern Shrubland (SS: 30.8°–40° N, 119.4°–105° W).

### 2.2  CMIP5 Models

All the Earth System Models (ESMs) in the CMIP5 archive that have soil moisture data are evaluated in this study. We evaluate monthly near-surface (0–10 cm) soil moisture from 17 ESMs and soil column (0–100 cm) soil moisture from 14 ESMs that are part of the CMIP5 archive (Table 1). Our analysis uses data from 2003 to 2012 because this is the time period with the greater number of in situ observations. Although the CMIP5 experiment ends in 2005, some ESMs, such as BCC-CSM1.1 and CanESM2, have an extended historical simulation through 2012. Therefore, we extend all the model simulations to 2012 by combining the 2006–2012 outputs from future emission scenario: the representative concentration pathways (RCP) 4.5 to the regular historical experiment outputs. RCP 4.5 is a pathway for stabilization of radiative forcing at 4.5 W m$^{-2}$ by 2100 (Thomson et al., 2011). A similar approach was adopted in the IPCC AR5 report (Bindoff et al., 2013). Jones et al. (2013) also used RCP4.5-forced CMIP5 simulations from 2005-2010 to investigate near-surface temperature variations. To validate this approach, we compared simulated precipitation based on different RCP scenarios with the Climatic Research Unit (CRU) precipitation in CONUS from 2006 to 2012 (results not shown) and found that the RCP 4.5 simulations closely match the CRU observations.

Because all of the ESMs have a different spatial resolution, model output is regridded to a uniform resolution of 0.25° × 0.25° using bilinear interpolation method. This resolution was chosen to match the spatial resolution of the satellite observations. Bilinear interpolation is a common method for interpolating precipitation (Chen and Frauenfeld, 2014;Hsu et al., 2013;Qu et al., 2013). Crow et al. (2012) demonstrated that large-scale spatial patterns of soil moisture are dominated by precipitation. Therefore, we believe that bilinear interpolation method is an appropriate method.

### 2.3  In Situ Observations

Daily in situ soil moisture from 2003 to 2012 were obtained from North American Soil Moisture Database (http://soilmoisture.tamu.edu/). The North American Soil Moisture Database archives data from a variety of national and





state networks (Quiring et al., 2016). Data from 363 stations are used in this study (Figure 1). These stations are collected from eight observational networks, as shown in Table 2. Quality-controlled daily soil moisture have been used to validate model simulations in previous studies (Xia et al., 2015c;Dirmeyer et al., 2016). In this study, any stations with short periods of missing data (<10 days) are infilled using the daily average replacement (DAR) method (Ford and Quiring, 2014b). Soil

moisture measurements at different depths are used to estimate the volumetric water content (VWC) in the top 10 cm and top 100 cm of the soil column. For example, the VWC measured at 5 cm is assumed to represent the VWC in 0–10 cm soil layer. When there are multiple soil moisture sensors within the top 100 cm, the measurements are combined using a depth-weighted average. Daily soil moisture measurements are then averaged to a monthly value to match the temporal resolution of the ESMs. The in situ measurements are also aggregated spatially to facilitate comparison with the CMIP5 models. We

use a simple spatial average to aggregate all of the stations within each 0.25° × 0.25° grid cell. Then all of the grid cells with stations in them are averaged to produce a regional or national dataset for comparing the in situ and modelled soil moisture. Although this spatial average method is not the optimal technique to reduce sampling errors (Crow et al., 2012), it is simple and has been widely used in previous model evaluations (Robock et al., 2003;Albergel et al., 2012a;Xia et al., 2015b). This approach reduces some of the bias associated with the point-versus-grid scale mismatch. Utilization of this approach over the

entire CONUS provides an overview of soil moisture simulations in CMIP5 models. However, we are also interested in spatial variations in model performance. Therefore, we also evaluated model performance after dividing CONUS into eight regions.

Measuring water content in frozen soils is a challenge (Xia et al., 2015c). Therefore, the CONUS analysis only evaluates the CMIP5 simulations during the warm season (April-September). For regional evaluation, we use data from all the months in

the three southern regions (Southeast, Southern Great Plains and Southern Shrubland) where frozen soils do not occur. All other regions only use data from the warm season.

## 2.4 Satellite Observations

Satellite-derived soil moisture from the soil moisture climate change initiative (CCI) project (http://www.esa-soilmoisture-cci.org/) is used in this study. This project is a part of the European Space Agency Programme on Global Monitoring of

Essential Climate Variables (ECV) (Liu et al., 2012). ECV soil moisture is based on active and passive remote sensing data and it has been validated using reanalyses (Albergel et al., 2013a;Albergel et al., 2013b) and in situ observations (Pratola et al., 2014). The spatial resolution of monthly ECV soil moisture is 0.25°. ECV soil moisture is not available during the cold season in the northern United States. Therefore, similar to the in situ observations, only warm season evaluations are undertaken for CONUS and the five northern regions. Data from all months is used in the three southern regions.

## 30 2.5 Evaluation Metric

Pearson correlation (r), mean absolute error (MAE), and the coefficient of efficiency (E) (Legates and McCabe, 1999) are used to quantify the agreement between observations and model simulations. Taylor's skill score (S) (Taylor, 2001) is also



used to measure the ability of individual CMIP5 models to reproduce the climatological soil moisture distribution. The equation of S is shown as following, Eq. (1):

$$S = \frac{4(1+R)}{\left[\sigma + (1/\sigma)\right]^2 (1+R_0)}$$ (1)

where $R$ is the correlation between the simulated and observed soil moisture. $\sigma$ is the ratio of standard deviation of model simulation over standard deviation of observation, and $R_0$ is the theoretical maximum correlation, equals to 1.

## 3 Results and Discussion

### 3.1 Evaluation of model ensemble over CONUS

Figure 2 shows the relationship between the CMIP5 ensemble mean and satellite-derived and in situ soil moisture during the warm season. All three of these datasets were averaged over CONUS. The multi-model ensemble mean is highly correlated with the in situ observations (Figure 2a and 2c). The correlation (r) between the in situ and model-derived soil moisture is 0.92 in the 0–10 cm soil layer and it is 0.91 in the 0–100 cm soil layer. Both of these correlations are statistically significant ($p < 0.05$). In the 0–100 cm soil layer, the CMIP5 soil water content is systematically higher than the in situ observations, especially during drier months (i.e., when soil water content is <0.25 cm$^3$ cm$^{-3}$). Figure 2b shows that there is a weaker relationship between the CMIP5 ensemble and ECV soil moisture and the correlation is only 0.65. It appears that the variance of the satellite-derived soil moisture is much less than the CMIP5 ensemble. The ECV soil moisture only varies from ~0.18 to 0.24 cm$^3$ cm$^{-3}$, while CMIP5 varies from ~0.16 to 0.27 cm$^3$ cm$^{-3}$. Therefore, the ECV soil moisture tends to be systematically greater than CMIP5 during drier months and systematically lower than CMIP5 during wetter months.

We also examined the mean monthly soil moisture in the 0–10 cm and 0–100 cm soil layers from April to September. Figure 3a shows the seasonal cycle in the 0–10 cm soil moisture for the in situ observations, ECV satellite data and CMIP5 models. Although there are substantial inter-model variations among the CMIP5 models, particularly with regards to the absolute soil water content, the CMIP5 ensemble (black line) shows strong agreement with in situ observations (red line). Both show that soil moisture decreases from April until August and then soil moisture recharge begins starting in September. Both the CMIP5 ensemble and the in situ observations have a similar seasonal cycle in terms of both the magnitude and timing. In comparison, the satellite-derived ECV soil moisture (blue line) shows little month to month variability and has a very weak seasonal cycle. Neither the timing nor the magnitude of these variations matches the in situ observations and the CMIP5 ensemble.

Figure 3b shows the seasonal cycle in the 0–100 cm soil moisture for the in situ observations and the CMIP5 models. ECV soil moisture data are not shown since satellites are only able to estimate near-surface soil moisture. The seasonal cycle of soil moisture in the 0–100 cm layer is similar to the 0–10 cm layer. Soil water content is highest during the early part of the





warm season (April/May) and it declines until reaching a minimum in August. There is general agreement between the in situ observations and CMIP5 ensemble, however there are notable differences in the magnitude of the soil water content. In addition, the dry down shown in the CMIP5 ensemble is less pronounced than in the in situ observations. There are substantial inter-model variations among the CMIP5 models, particularly with regards to the absolute soil water content which is similar to the 0–10 cm soil layer. We will focus on evaluating the performance of individual models in the following sections of the paper.

We also compared the spatial pattern of the mean soil moisture (2003–2012) during the warm season (April–September) (Figure 4). Based on the CMIP5 ensemble, the soils with the lowest soil water content in the 0–10 cm layer are typically found in the southwestern U.S. and the soils with the highest soil water content tend to be found in the northeastern U.S. (Figure 4a). This pattern is also evident in the 0–100 cm soil layer, however the gradient is less pronounced (Figure 4b). The patterns are somewhat less spatially consistent when one examines the in situ observations because of the influence of local factors (e.g., edaphic, climatic, topographic, vegetation, etc.).

The differences between CMIP5 and the in situ observations are shown in Figure 4e and 4f. Generally, CMIP5 tends to be significantly wetter than the in situ observations in the western U.S. and it tends to be significantly drier than the in situ observations in the eastern U.S. In fact, 79.3 percent of the differences between CMIP5 and the in situ observations in the 0–10 cm layer are statistically significant. The same patterns are evident in the differences between CMIP5 and the in situ observations in the 0–100 cm layer (Figure 4f). However, a greater number (8.8% more) of the positive biases in western U.S. and the negative biases in the eastern U.S. are statistically significant than in the 0–10 cm layer. These results agree with previous research. Sheffield et al. (2013) concluded that the CMIP5 models tend to overestimate precipitation in west North America. Given that precipitation is a principal control of soil moisture, a positive bias in precipitation can cause soils to be too wet. Sheffield et al. (2013) also found that CMIP5 models tend to overestimate evaporation in eastern North America. This would lead to drier soils and could help to explain the dry biases in CMIP5 that were observed in the eastern U.S.

Figure 5 compares the mean warm season (April–September) soil moisture (2003–2012) in 0–10 cm soil layer from the CMIP5 ensemble to the satellite-derived ECV soil moisture. The general spatial pattern of ECV is consistent with CMIP5, however ECV has much great spatial heterogeneity. This is partly due to the finer spatial resolution of the ECV data as compared to CMIP5. It is also apparent that there are significant differences in the near-surface soil water content in ECV versus CMIP5. For example, ECV shows that the regions with relatively low soil water content during the warm season (VWC < 0.2) are much more spatially extensive than in CMIP5. Similarly, the areas with relatively high soil water content (VWC > 0.3) are also more extensive with EVC. There has also been a shift in the soil water maxima in ECV into Maine and New Hampshire, with secondary maxima in Washington. The spatial pattern of the differences between ECV and CMIP5 are similar to those seen with the in situ observations. CMIP5 tends to have wet biases in the western U.S. and dry biases in the eastern U.S. The majority of statistically significant (p < 0.05) differences are concentrated in the places where CMIP5 is wetter than ECV (regions with a wet bias).




We evaluate the performance of each CMIP5 model over CONUS during the warm season using Taylor's skill score, as shown in Figure 6. Based on the skill score, the individual models show a varying ability to capture the soil moisture distribution over CONUS. In the 0–10 cm soil layer, CCSM4, NorESM1-M, CESM1 and GFDL-ESM2M all perform well (when compared to in situ observations) and have higher skill scores (S = 0.89, 0.87, 0.87 and 0.85) than the CMIP5

ensemble (S = 0.84). CanESM2 (S = 0.39), INM-CM4 (S = 0.47) and HadGEM2-ES (S = 0.46) have the lowest scores.

When model performance is evaluated using ECV soil moisture, the skill scores decrease for all the models. Among the 17 CMIP5 models that were evaluated, 8 have higher skill scores than the CMIP5 ensemble mean (BCC-CSM1.1, CCSM4, CESM1, FGOALS-g2, GFDL-ESM2M, GISS-E2-H, IPSL-CM5A-LR and MIROC-ESM). In the 0–100 cm soil layer, CCSM4 (S = 0.86), CESM1 (S = 0.88), GFDL-ESM2M (S = 0.80) and HadGEM2-ES (S = 0.89) perform well. The

performance of CanESM2, INM-CM4 and HadGEM2-ES improves in this layer as compared to the 0–10 cm layer. Generally, CCSM4, CESM1 and GFDL-ESM2M consistently perform well over CONUS in both the near-surface and deeper soil layers.

The performance of each CMIP5 model is also evaluated using correlation, RMSE and "amplitude of variations" (relative standard deviation). These metrics are represented in Figure 7 using a Taylor diagram (Taylor, 2001). Correlations between

soil moisture simulated by CMIP5 models and ECV and in situ observations are indicated by the azimuthal position of each dot in Figure 7. Correlations (r) between simulated 0–10 cm soil moisture and ECV observations (Figure 7a) are all lower than 0.7. They tend to be clustered around 0.6, with the exception of BNU_ESM. Correlations between the CMIP5 models and the in situ soil moisture observations are more variable, as shown in Figure 7b. CCSM4 and CESM1 (r = 0.79) have the highest correlations, while IPSL-CM5A-LR (r = 0.55) and GISS-E2-H (r = 0.56) have the lowest correlations. The radial

distance from the origin represents the standardized deviation of the CMIP5 models relative to the standardized deviation of the observations. When examing the performance of the CMIP5 models in the 0–10 cm soil layer, CanESM2, INM-CM4 and HadGEM2-ES are outliers showing much larger ($\sigma_{sim} / \sigma_{obs} > 2$) variations than either ECV or in situ observations. This leads to low Taylor's skill scores for these three models. All the models show larger variations than ECV soil moisture, while only 10 (out of 17) models demonstrate larger variations than in situ soil moisture. In the 0–100 cm soil layer, the models in

Figure 7c are more clustered than in 0–10 cm soil layer. In general, the models tend to under-estimate the variability in the 0–100 cm layer. 12 of the 14 models have standardized deviations that are lower than the observations. This indicates that most of the models cannot capture the true variability of soil moisture in this layer. INM-CM4 significantly overestimates the standardized deviation which is consistent with the results for the 0–10 cm soil layer. FGOALS-g2 (S = 0.56) has the lowest Taylor's skill score in the 0–100 cm layer. This is due to the low correlation (r = 0.69) and the model also

significantly underestimates soil moisture variability ($\sigma_{sim} / \sigma_{obs} = 0.51$).

## 3.2 Regional Evaluation

The CMIP5 models are also evaluated in eight regions in CONUS (Figure 8). Correlations between model-simulated and in situ surface soil moisture (green bar) are higher in all regions than the correlations (blue bar) based on ECV soil moisture,



except in the NGP region (Figure 8a). Focusing on the correlations between CMIP5 ensemble and in situ soil moisture, correlations for 0–100 cm soil moisture (brown bar) are similar to the correlations for 0–10 cm soil moisture. Only in the NGP region, correlation in 0–100 cm soil layer is substantially higher than in 0–10 cm soil layer. Examining the MAE gives a different perspective. In most regions, the CMIP5 ensemble has a lower MAE when compared to ECV versus the in situ

observations. Only in the Northern Shrubland and Southern Shrubland regions is the MAE lower when computer to the in situ observations. Figure 8b indicates that MAE in the 0–100 cm soil layer is substantially higher than MAE in 0–10 cm soil layer in 7 of the 8 regions. Similarly, the coefficient of efficiency is generally higher in the 0–10 cm layer than in 0–100 cm. Model performance varies from region to region. Based on the ECV soil moisture, the CMIP5 ensemble has relatively high correlations (r = 0.64 and 0.66) in the MW and NE and relatively low correlations (r = 0.23) in the SS region. Based on the

in situ soil moisture, correlations are consistently high (r > 0.85) in NS, NW, SE, SGP and SS in both the near-surface and deep soil layers. The lowest correlation (r = 0.50) between the near-surface in situ soil moisture and CMIP5 ensemble is in the NGP. The MAE based on ECV soil moisture is relatively low in the NE, NGP and NW (MAE = 0.021, 0.021 and 0.021 $cm^3$ $cm^{-3}$) and relatively high in NS and SS (MAE = 0.042 and 0.046 $cm^3$ $cm^{-3}$). However, when compared to the near-surface in situ soil moisture, MAE is relatively high in the NW (MAE = 0.037 $cm^3$ $cm^{-3}$).

There is substantially more regional variability in MAE for the 0–100 cm soil moisture. The MAE exceeds 0.07 $cm^3$ $cm^{-3}$ in NS and NW, while in the NGP it is only 0.03 $cm^3$ $cm^{-3}$. The regional variation in coefficient of efficiency (E) is also substantial. When E is calculated based on the in situ observations it demonstrates that the CMIP5 ensemble can skillfully simulate the 0–10 cm soil moisture in the NS, NW, SE, SGP and SS regions. The results also demonstrate that CMIP5 can accurately simulate the 0–100 cm soil moisture in the NS, SE, SGP and SS regions during the warm season. However, these

results do not agree with the performance assessment based on the ECV soil moisture. Based on ECV, E is best in the MW and NE regions and CMIP5 model ensemble is worse than climatology in the SS region.

Based on the results presented above, model performance differs significantly when being evaluated with in situ versus ECV soil moisture. In addition, the selection of the best performing models is dependent on which statistic is used. For example, based on the in situ soil moisture in 0–10 cm layer, the NS and NW regions have relatively high MAE (MAE = 0.037 and

0.037 $cm^3$ $cm^{-3}$) even though the correlations are also strong (r = 0.87 and 0.89). This suggests that the model is able to simulate the wetting and drying of the soil, but there is a systematic bias in the absolute magnitude of the model-simulated soil moisture.

Figure 9 shows the skill scores of each model in the eight regions using in situ observations from the warm season as reference. There is substantial inter-model variability in performance amongst the CMIP5 models as a function of soil depth

and location. CESM1 has consistently high skill in the 0–10 cm soil layer in all eight regions. MRI-CGCM3 outperforms all the other models in the MW region and it also performs well in the NE along with ACCESS1.3. CanESM2 and HadGEM2-ES do not perform well in the majority of regions (6 out of 8 regions) and GISS-E2-H does not perform well in the MW and NE. For the 0–100 cm soil layer, HadGEM2-ES performs well in all regions, especially in NGP, NS, NW, SE and SGP. The





models generally perform better in the NE, compared to other regions. FGOALS-g2 and GISS-E2-H perform relatively poorly in all regions.

Due to the availability of ECV data and the issues with measuring soil moisture in frozen soils, the preceding analysis focused solely on the warm season. We also evaluated model performance using data from all months in the three southern regions (SE, SGP and SS) where frozen soils are not an issue. Figure 10 shows the seasonal cycle of soil moisture based on the CMIP5 ensemble, in situ and ECV data in the three southern regions. CMIP5 ensembles in the three regions consistently show that soil moisture decreases first then increases in a year. However in SE, soil moisture reaches driest condition (in September) later than soil moisture in SGP (August) and SS (July). Both the in situ and ECV show more variable seasonal patterns than the CMIP5 simulations, especially in the SGP and SS. In the SE, both the in situ and ECV soil moisture decrease starting in February and reach their lowest point in June. This is three months earlier than the CMIP5 ensemble. In situ observations are wetter than ECV soil moisture during the entire year in the SE, but they are most similar in October. In the SGP, in situ and ECV soil moisture generally decreases from April to August and then increases after August. There is good agreement between the in situ, ECV and CMIP5 in the SGP with regards to the timing of the wettest and driest months. This is the only region where the seasonal cycle is the same in all three data sources. However, the magnitude of the seasonal fluctuations differs substantially. CMIP5 is much more variable than both the in situ and ECV. While in the SS region, the ECV does not show much of a seasonal cycle. CMIP5 and the in situ observations show a similar drying of the soil from March through June, but they do not agree as well during the June to November period. Table 3 provides the correlation, MAE and E based on the month data from these three regions. During the warm season months the correlations and coefficient of efficiency are higher and the MAE is lower in all the cases. In terms of the surface layer, the CMIP5 ensemble is more highly correlated with in situ observations than ECV data in all three regions. However, in the SGP and SE, the MAE based on comparing the CMIP5 ensemble to the ECV is lower than the MAE based on the in situ observations. With emphasis on in situ soil moisture in different layers, CMIP5 ensemble has higher correlation, larger MAE and lower E in 0–100 cm soil layer than in 0–10 cm soil layer in all the three regions.

## 4    Limitations

This study compares model-simulated soil moisture from the CMIP5 models with in situ and satellite-derived soil moisture. The in situ stations were selected based on their record length spatial coverage. However, there are relatively few stations with 10-year records. Therefore, some parts of CONUS are not well represented in this analysis. Future studies would benefit from including more in situ data to evaluate model performance. This would help to address issues with the spatial gaps in coverage and the issues related to comparing point measurements to model grid cells. Considering the in situ soil moisture come from different networks, there may also be some inconsistencies in the quality and representativeness of the soil moisture data (Dirmeyer et al., 2016). These inconsistencies can result from the use of different soil moisture sensors, calibration procedures and quality control processes. Dirmeyer et al. (2016) assessed the random errors of 16 networks and





found distinct differences between networks. Although we excluded from this study one of the networks with the largest random errors (e.g., COSMOS), more work is still needed to standardize and homogenize in situ soil moisture measurements. Another potential limitation of this work is that we applied bilinear interpolation method to regrid all the CMIP5 model output to a uniform resolution of 0.25° × 0.25° so that it matched the resolution of the ECV data. This is a simple way of re-

scaling the data. Given that we are only evaluating model performance at the regional and continental scale, we believe that this method is reasonable because the spatial variability of soil moisture at these scales is dominated by precipitation patterns (Crow et al., 2012). However, applying more advanced interpolation or downscaling methods such as the reduced optimal interpolation (ROI) method (Yuan and Quiring, 2016) may provide a better estimates of model-simulated soil moisture at this spatial scale.

## 10  5  Conclusions

We evaluated soil moisture simulations in CMIP5 experiment (17 models for 0–10 cm and 14 models for 0–100 cm) over CONUS using in situ observations and ECV satellite observations. The CONUS results show that the CMIP5 model ensemble has similar correlations with in situ observations when comparing the 0–100 cm soil layer with the 0–10 cm soil layer. However, there is evidence of a substantial wet bias in the deeper soil layer during months when the soil is dry. This

wet bias is also reflected in the multi-year mean monthly soil moisture. There is substantial variability in performance among the individual models, with the greater uncertainties in surface soil layer.

The multi-model CMIP5 ensemble mean can generally capture the spatial pattern of soil moisture. However, wet biases in the western U.S. and dry biases in the eastern U.S. are evident. Sheffield et al. (2013) found that CMIP5 models tend to overestimate precipitation in the western U.S. and this may account for the wet biases that we observed. Dry biases in the

eastern U.S. may be attributed to evapotranspiration, which tends to overestimated by CMIP5 models in the eastern U.S. (Sheffield et al., 2013). Performance of the CMIP5 ensemble varies significantly from region to region. In most regions (NS, NW, SE, SGP and SS), the CMIP5 ensemble can accurately simulate warm season surface soil moisture (e.g., high correlations and low MAE). In the three southern regions, we also evaluated soil moisture simulations during the cold season and found that there is generally a decrease in model performance (e.g., higher MAE and lower E than during the warm

season).

ECV soil moisture, as an independent data source, is introduced in this study to help evaluate the performance of CMIP5 soil moisture simulations. Relative to ECV soil moisture, CMIP5 ensemble shows greater month-to-month variations over CONUS. Due to this greater variance, CMIP5 models do not skillfully reproduce the ECV soil moisture. Similar with in situ soil moisture, ECV data also shows that the CMIP5 model ensemble tends to have wet biases in the western U.S. and dry

biases in the eastern U.S. Additionally, in the three southern regions, the intra-annual variability shown by ECV soil moisture and in situ observations are relatively consistent. On the other hand, the CMIP5 ensemble can only capture the general seasonal cycle, but fails to adequately capture some of the monthly variations. At the same time, there are some





inconsistencies between the in situ and ECV soil moisture. For example, in the Southern Shrubland, the correlation between the CMIP5 models and ECV soil moisture (r = 0.23) is lower than the correlation with the in situ data (r = 0.87). Though comparing the two observational data is not the goal of this study, we can still point out future validation of satellite derived soil moisture is necessary.

The skill of the individual CMIP5 models also varies significantly. In the top soil layer, the Taylor skill score varies from 0.39 (CanESM2) to 0.89 (CCSM4). Generally, the skill of the models in the deeper soil layer is similar to the surface layer, but the inter-model variability in skill is greater. HadGEM2-ES has the highest skill score because it matches the variability of the in situ observations. Generally, CESM1 consistently performs well in the surface soil layer in all regions, and HadGEM2-ES performs well in the 0–100 cm soil layers in all regions. However, it is remains difficult to find a single

model that consistently outperforms all others when it comes to accurately simulating soil moisture in all regions and seasons. Therefore, it is unclear whether the findings of this study will apply to other regions around the world with difference climate, soil and vegetation characteristics.

**Acknowledgments**

This work was supported by NSF Grant AGS-1056796 to Texas A&M University. The in situ soil moisture data set is available at TAMU North American Soil Moisture Database (http://soilmoisture.tamu.edu/). The satellite observed soil moisture is available at the the European Space Agency soil moisture climate change initiative (CCI) project website (http://www.esa-soilmoisture-cci.org/). CMIP5 simulated soil moisture is served by the Earth System Grid Federation (ESGF)

(https://pcmdi9.llnl.gov/).

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





**Table 1. List of 17 CMIP5 Models in This Study**

| Model Name | Model Center (or Groups) | Spatial Resolution | Soil Moisture Simulation | |
|---|---|---|---|---|
| | | | 0-10cm | 0-100 cm |
| ACCESS1.3 | Commonwealth Scientific and Industrial Research Organization (CSIRO) and Bureau of Meteorology (BOM), Australia | 145×192 | √ | √ |
| BCC-CSM1.1 | Beijing Climate Center, China Meteorological Administration | 64×128 | √ | √ |
| BNU-ESM | College of Global Change and Earth System Science, Beijing Normal University | 64×128 | √ | √ |
| CanESM2 | Canadian Centre for Climate Modelling and Analysis | 64×128 | √ | √ |
| CCSM4 | National Center for Atmospheric Research | 192×228 | √ | √ |
| CESM1(CAM5) | Community Earth System Model Contributors | 192×228 | √ | √ |
| CNRM-CM5 | Centre National de Recherches Météorologiques and Centre Européen de Recherche et Formation Avancée en Calcul Scientifique | 192×228 | √ | √ |
| CSIRO-MK3.6.0 | Commonwealth Scientific and Industrial Research Organization in collaboration with Queensland Climate Change Centre of Excellence | 96×192 | √ | |
| FGOALS-g2 | LASG, Institute of Atmospheric Physics, Chinese Academy of Sciences | 60×128 | √ | √ |
| GFDL-ESM2M | NOAA Geophysical Fluid Dynamics Laboratory | 90×144 | √ | √ |
| GISS-E2-H | NASA Goddard Institute for Space Studies | 90×144 | √ | √ |
| HadGEM2-ES | Met Office Hadley Centre (additional realizations contributed by Instituto Nacional de Pesquisas Espaciais) | 145×192 | √ | √ |
| INM-CM4 | Institute for Numerical Mathematics | 120×180 | √ | √ |
| IPSL-CM5A-LR | Institut Pierre-Simon Laplace | 96×96 | √ | |
| MIROC-ESM | Japan Agency for Marine-Earth Science and Technology Atmosphere and Ocean Research Institute (The University of Tokyo) and National Institute for Environmental Studies | 64×128 | √ | √ |
| MRI-CGCM3 | Meteorological Research Institute | 160×320 | √ | |
| NorESM1-M | Norwegian Climate Centre | 96×144 | √ | √ |



**Table 2. List of Observational Networks in This Study**

| Network | Number of Sites (Used in this study) | Reference |
|---|---|---|
| AmeriFlux | 4 | (Baldocchi et al., 2001) |
| North Carolina Environment and Climate Observing Network | 24 | (Pan et al., 2012) |
| Illinois Climate Network | 16 | (Hollinger et al., 1994) |
| Michigan Automated Weather Network | 34 | (Andresen et al., 2011) |
| Oklahoma Mesonet | 104 | (Scott et al., 2013) |
| Soil Climate Analysis Network | 66 | (Schaefer et al., 2007) |
| Snowpack Telemetry | 97 | (Schaefer and Paetzold, 2001) |
| West Texas Mesonet | 18 | (Schroeder et al., 2005) |





**Table 3. Evaluation of CMIP5 Ensemble over Southeast, Southern Great Plains and Southern Shrubland Using All Monthly Soil Moisture and Warm Season Only Soil Moisture**

| | | Correlation | | MAE | | E | |
|---|---|---|---|---|---|---|---|
| | | All | Warm | All | Warm | All | Warm |
| SE | v.s. ECV | 0.44 | 0.50 | 0.030 | 0.024 | 0.11 | 0.17 |
| | v.s. In Situ (0 - 10 cm) | 0.80 | 0.88 | 0.032 | 0.028 | 0.61 | 0.72 |
| | v.s. In Situ (0 - 100 cm) | 0.89 | 0.91 | 0.067 | 0.052 | 0.21 | 0.43 |
| SGP | v.s. ECV | 0.38 | 0.44 | 0.026 | 0.024 | 0.05 | 0.15 |
| | v.s. In Situ (0 - 10 cm) | 0.82 | 0.86 | 0.032 | 0.027 | 0.63 | 0.67 |
| | v.s. In Situ (0 - 100 cm) | 0.90 | 0.92 | 0.071 | 0.056 | 0.19 | 0.45 |
| SS | v.s. ECV | 0.21 | 0.23 | 0.051 | 0.046 | -1.12 | -0.76 |
| | v.s. In Situ (0 - 10 cm) | 0.81 | 0.87 | 0.028 | 0.023 | 0.66 | 0.73 |
| | v.s. In Situ (0 - 100 cm) | 0.88 | 0.89 | 0.074 | 0.055 | 0.17 | 0.41 |





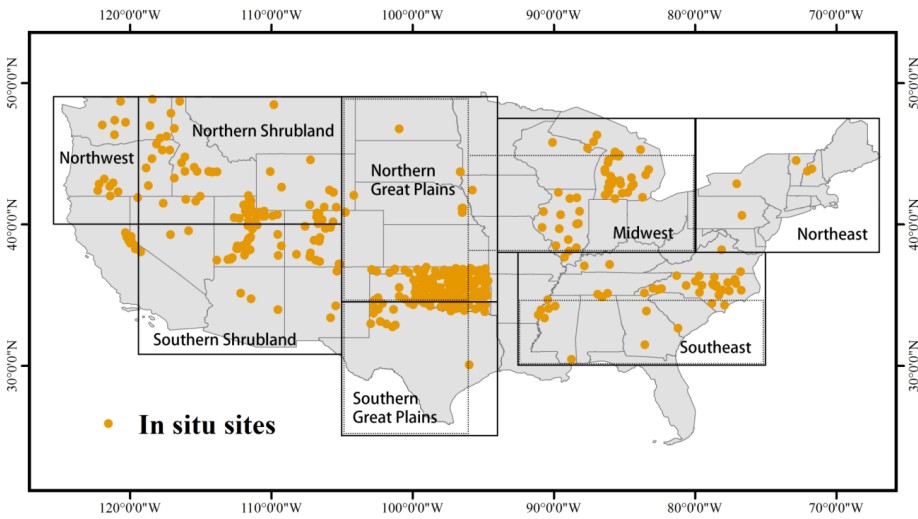

**Figure 1: Spatial distribution of in situ soil moisture stations and the boundary of the 8 sub-regions.**





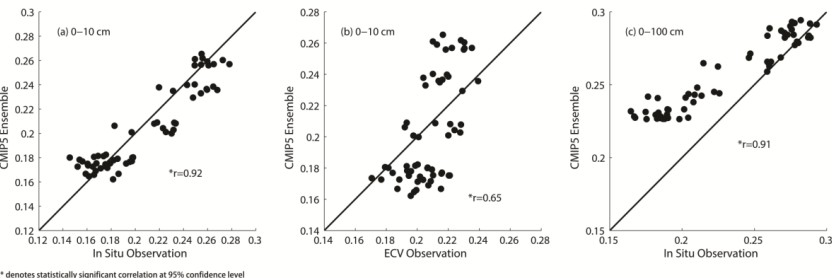

**Figure 2. Comparison of soil moisture from the CMIP5 ensemble with in situ and satellite-derived (ECV) soil moisture. Each point represents monthly soil moisture data from the warm season (April to September) that has been spatially-averaged over CONUS (2003-2012). (a) CMIP5 ensemble versus in situ observations in the 0–10 cm soil layer. (b) CMIP5 ensemble versus ECV in the 0–10 cm soil layer. (c) CMIP5 ensemble versus in situ observations in the 0–100 cm soil layer during warm season (April to September).**

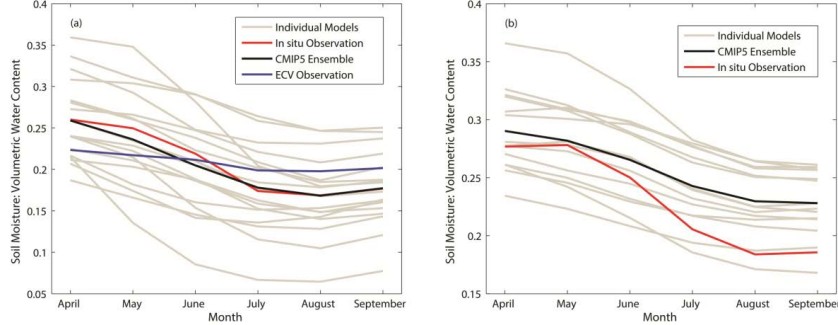

**Figure 3. Mean monthly soil moisture (2003-2012) during the warm season (April to September) in the 0–10 cm soil layer (a) and in the 0–100 cm soil layer (b). Data are spatially-averaged over CONUS. Figures show the monthly mean soil moisture from the in situ observations (red line), ECV satellite data (blue line), CMIP5 ensemble (black line) and the individual CMIP5 models (grey lines).**





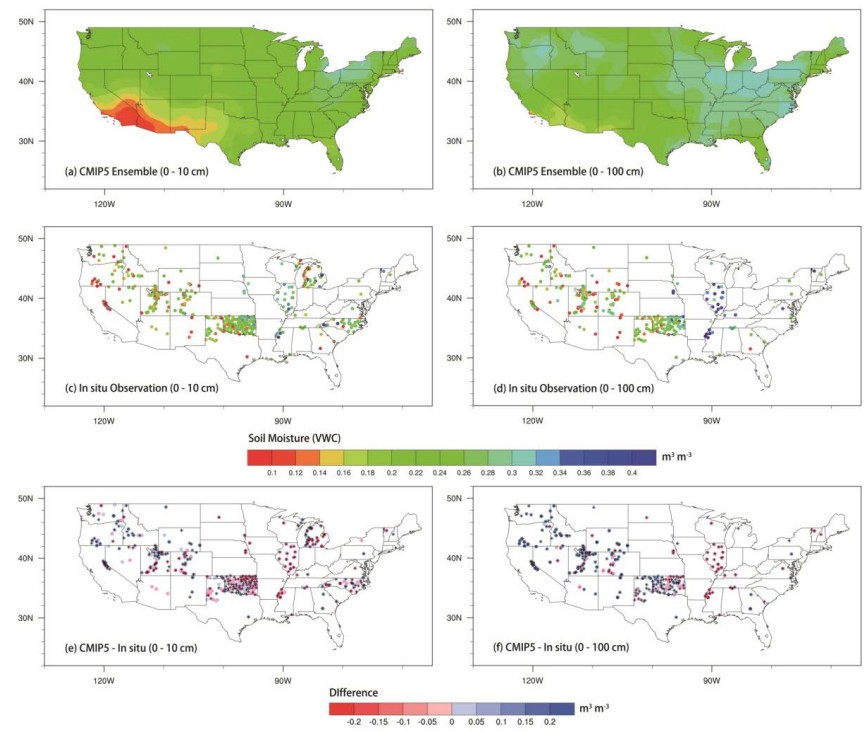

**Figure 4. Mean soil moisture (m³ m⁻³) over CONUS (2003–2012) during warm season (April to September). Left panel: 0–10 cm soil moisture for: (a) CMIP5 ensemble, (c) in situ observations and (e) the difference between them (CMIP5 – in situ). Right panel: 0–100 cm soil moisture for (b) CMIP5 ensemble, (d) in situ observations and (f) the difference between them (CMIP5 – in situ).**





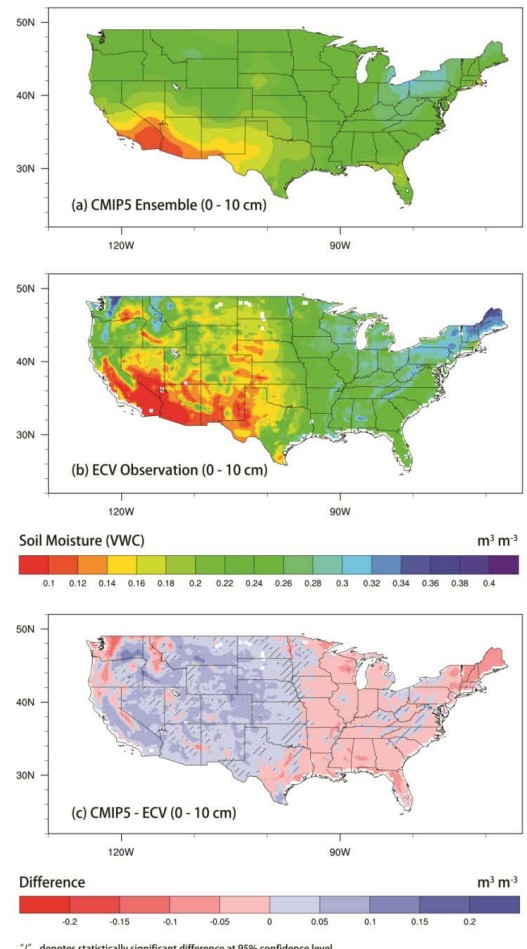

**Figure 5. Same as Figure 4, except it compares the mean soil moisture (m³ m⁻³) over CONUS (2003–2012) during warm season (April to September) from the CMIP5 ensemble with the satellite-derived ECV soil moisture.**




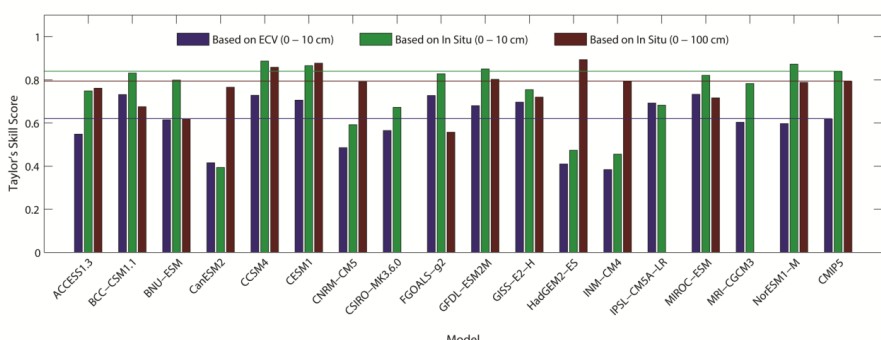

**Figure 6. Taylor skill scores of CMIP5 over CONUS based on the ECV satellite data (blue) and in situ observations in the 0–10 cm soil layer (green) and in 0–100 cm soil moisture (brown). The solid lines indicate the skill of the CMIP5 ensemble average.**

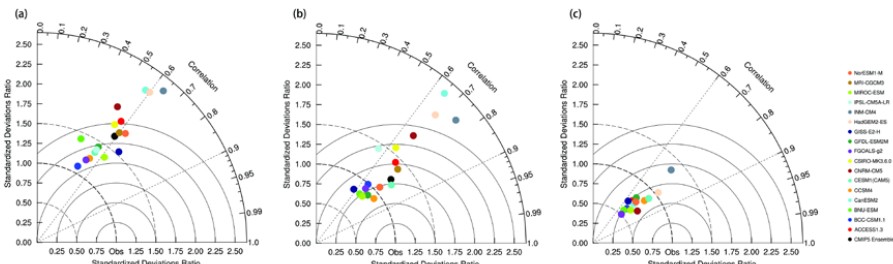

**Figure 7. Taylor diagrams for the CMIP5 models based on the (a) ECV satellite data (b) in situ observations in the 0 - 10 cm layer and (c) in situ observations in the 0–100 cm layer. Azimuthal angle represents correlation coefficient and radial distance is the standard deviation normalized to observations.**

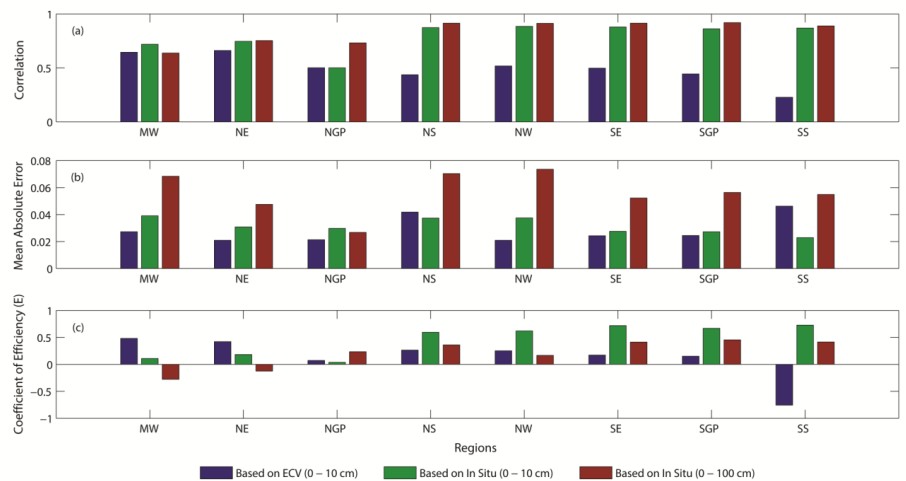

**Figure 8. Performance evaluation statistics for CMIP5 ensemble mean versus ECV satellite data and in situ soil moisture (2003–2012): (a) correlation coefficient, (b) mean absolute error, and (c) coefficient of efficiency for the eight regions.**





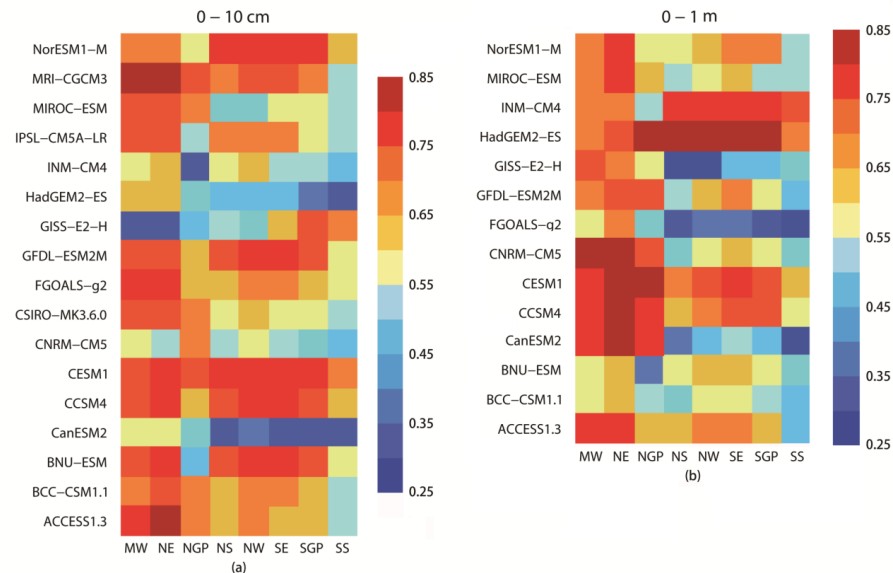

Figure 9. Comparison of CMIP5 models with in situ observations over eight regions based on Taylor's skill scores: (a) 0–10 cm soil moisture, and (b) 0–100 cm soil moisture.

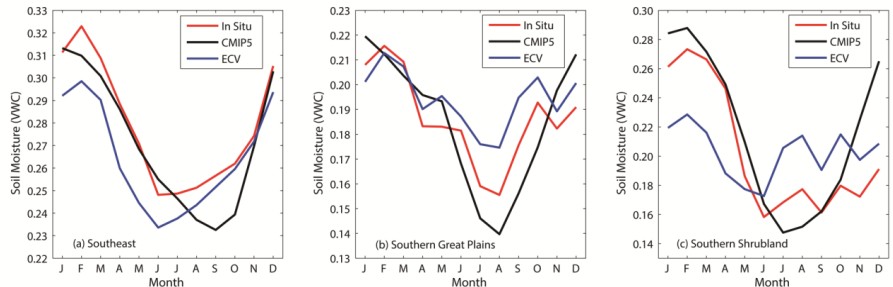

Figure 10. Seasonal variation of mean monthly (2003–2012) soil moisture based on in situ observations (red), CMIP5 ensemble
10   (black) and ECV satellite data (blue) in three regions: (a) Southeast, (b) Southern Great Plains, and (c) Southern Shrubland.