# Peer review of "Evaluation of soil moisture in CMIP5 simulations over the contiguous United States using in situ and satellite observations"

_Hydrology and Earth System Sciences, 2016_

## Referee Comment (RC1) · Anonymous Referee #1 · 27 Dec 2016

General Comments: This paper evaluated soil moisture simulations in CMIP5 experiment using in situ and satellite observations. The evaluation focused on both surface and deep soil layers. This paper clearly stated the research question, used defendable methodology and datasets and presented solid results and conclusions. This paper is logically formatted and set a foundation for using in situ observation to evaluate soil moisture from GCMs. Though there are still some limitations in this study (as the authors described in the last section), this paper can still bring benefits to future research and applications, such as model development and validation, drought evaluations and data assimilations. Hence, I recommend this paper to be published in HESS with minor

revision. Some detailed comments are attached as following.

Detailed Comments: 1. Section 2.1, page 4. The authors modified sub-regions from previous studies. Will this modification affect the results? For instance, do the land cover types in the new sub-regions differ from previous studies? 2. Page 5, line 2. The authors mentioned that soil moisture data were collected from 8 different networks. Do the 8 networks use same way to measure soil moisture? If no, then is there any significant biases among networks? 3. Comparison between point measurements and gridded value is a big challenge, especially in a big grid box. Can simple spatial average method solve the issue? 4. Page 6, line 14. Add a space between "<" and "0.25". 5. Section 3.1, page 8. The content in this page is about the evaluation of individual models. Generate another section to present these results. 6. Page 27, Figure 9b. It is better to change 1 m to 100 cm at the top of figure, because it is important to keep expressions consistent throughout the paper.

---

## Referee Comment (RC2) · R. Orth (Referee) · 9 Jan 2017

R. Orth (Referee)

rene.orth@env.ethz.ch

Review of Yuan and Quiring "Evaluation of soil moisture in CMIP5 simulations over contiguous United States using in situ and satellite observations"

This article is on the validation of the performance of CMIP5 models over the continental US in terms of the simulated soil moisture, using in situ and satellite-derived soil moisture data as observational reference. While the agreement of the multi-model mean with the observational references is overall satisfactory, the paper reveals strong inter-model differences, general difficulties of the models to capture the observed spatial pattern of absolute soil moisture, but also differences between in situ and satellite-based soil moisture.
* * *
General comments:

The article adresses an important topic and contributes valuable results, and should be of interest for the HESS audience. It helps to guide Earth System Model development, as well as the development of large-scale observational soil moisture products.

However, I do have several concerns:

(1) Both, the spatial downscaling of the CMIP5 output, and the gap filling of the in situ soil moisture records, are - in my view - unnecessary data manipulations which could introduce errroneous signals to the raw data. Instead of the downscaling of the model data, I would recommend to upscale the observational data to the coarse spatial resolution of the CMIP5 models, especially because you only focus at the regional-continental scale. Instead of the gap filling, I would compute the models' monthly averages using only the same days available in the observations. In any case I would expect some analyses indicating the impact of any data pre-processing you perform on the final results.

Also the satellite-based soil moisture data might not be available anywhere and any-time. The current manuscript does not mention how the authors deal with this.

(2) The comparison between the absolute soil moisture in the ECV data and the CMIP5 models is maybe not appropriate. I think the absolute soil moisture amount in the ECV data has been scaled using data from land surface model simulations, while only the temporal variations are a truly observed feature. This would mean that when comparing the spatial patterns of the absolute soil moisture contents you actually compare model against model.

(3) I think the results of this study can be very useful to guide model development, as

well as the development of large-scale observational soil moisture products. While I recognize that this is not the main goal of this study, I would like to see some more explanations why poor model performance or differences across in situ and satellite-based soil moisture are seen at several of the performed comparisons. This could then lead into explicit advice for the developers of the models and the soil moisture products.

(4) The results section can be significantly shortened. Description of results displayed in figures does not need to be so comprehensive.

(5) While the manuscript is clearly structured and overall easy to read, there are many small language errors (such as missing articles). I recommend that the authors take special care of these when revising the manuscript.

I recommend publication of this study after major revisions.

I do not wish to remain anonymous - Rene Orth.

————————————————

Specific comments:

Title:

... over the contiguous United States...

Abstract:

line 15: maybe replace 'magnitude' with 'amount' throught the manuscript

line 16: 'variations in model performance' could be spatial, temporal, or across models (which is what you mean, I guess). Please clarify.

line 16: 'especially in the near-surface', please replace with 'at' or 'for the near surface'

line 17: deeper soil layers

page 2:

[Figure]

line 5: please explain 'ground fluxes'

line 8: remote sensing observations

line 21: change 'predict ... earlier in ...' to 'better predict'

line 23: 'from satellites remote sensing', improve phrasing

line 30: Why is it a problem that ASCAT soil moisture is influenced by precipitation and evaporation? Soil moisture is by definition influenced by these quantities.

line 32: abbreviation AMSR-E not introduced

page 3:

line 7: 'offline land surface models', please improve phrasing

line 8: at a variety of depths

line 9: biases

line 11: please currect citation style

line 22: difficulties to accurately simulate

line 28: abbreviation CONUS not introduced

line 31: ... followed by the presentation of the results and a discussion in section 3. Limitations and conclusions of the study are ...

page 4:

section 2.1: How do the adjustments of the region perimeters influence the results of the study?

section 2.2: Why was this particular emission scenario used here?

line 3: please refer to 'eight sub-regions' instead of 'regions' throught the manuscript (as also CONUS is a region)

lines 7-10: a table would be suitable to present these information;

line 12: provide soil moisture data

line 15: 'with the greater number of in situ observations', please improve phrasing

lines 16-18: please improve phrasing

line 25: using a bilinear interpolation method

line 30: Daily in situ soil moisture data from ...

page 5:

section 2.3: Do your results depend on the amount of stations in each sub-region?

section 2.4: What version of the ESA CCI soil moisture is employed? Maybe an upgrade (if possible) would improve the coherence between the in situ and satellite-derived soil moisture results?

line 1-2: These stations belong to eight ...

line 2: Quality-controlled daily soil moisture data have ...

line 4: gap filling of missing data: Beside the main comment above, how does it work?

line 15: 'provides an overview of soil moisture simulations in CMIP5 models'?

line 19: For a regional evaluation...

page 6:

line 22: remove 'starting'

page 7:

line 1: agreement in terms of what?

line 5: 'which is similar to the 0-10 cm soil layer', please improve phrasing, maybe use

'Similar results are found for the 0-10 cm soil layer.'

line 18: and of the negative biases

line 26: ECV shows more spatial heterogeneity

line 34: '(regions with a wet bias)'? remove?

page 8:

line 25: in the 0-10 cm soil layer

page 9:

line 5: please rephrase

page 10:

line 7: the driest conditions

line 20: is more strongly correlated

page 11:

line 14: comparatively dry 'substantial bias in the deeper soil layer', can you speculate why that is?

line 17: the observed spatial pattern

line 21: varies significantly across sub-regions

line 27: the CMIP5 ensemble

line 31: 'relatively consistent', not in the SS sub-region

page 12:

line 3: point out that

Figure 1, caption: and the boundaries of

Figure 2: Please use the same x and y-axes in all plots.

Figure 3, caption: CMIP5 ensemble mean (black line)

Figure 4: Maybe add white color in the middle of the color bar such that locations with good agreement do not show up?

Figure 9: Please label color bars. I find it interesting that the correlations for the SS sub-region are consistently low, whereas in Figure 8 the correlations for the model ensemble mean in that sub-region are high. Can you comment on that?

―――――――――――――――

---

## Author Comment (AC1) · 14 Feb 2017

RESPONSE TO THE REVIEWER #1'S COMMENTS

We appreciate the reviewer's encouraging comments, and agree with the suggestions. Your comments will improve the manuscript. In accordance with these suggestions, we have revised the manuscript carefully. Responses to each comment are provided below.

Detailed comments

1. Section 2.1, page 4. The authors modified sub-regions from previous studies. Will this modification affect the results? For instance, do the land cover types in the new sub-regions differ from previous studies?

Response:

Thanks for this question. We have generally used the same sub-regions as in previous studies, but four sub-regions (Northern Great Plains, Southern Great Plains, Midwest and Southeast) were modified so that we could include more in situ soil moisture measurements. The original boundaries of each sub-region were based on the land cover types and were applied in soil moisture related research (Mei and Wang, 2012). We found the land cover types in the modified sub-regions do not change greatly. In Northern and Southern Great Plains, the main effect of the modification is including more soil moisture sites in eastern Oklahoma where the dominate land cover type is Savanna. Savanna is also the main land cover type in central Oklahoma. In the Midwest, the modified region expands northward. Most of the sites (5 of 6) that were added are located in cropland region. This is consistent with the dominant land cover, since cropland covers more than 90% area of the Mideast. In the original Southeast, evergreen forest is the main land cover type. The land cover types in the modified Southeast is mixed by evergreen forest, deciduous forest and mixed forest. This is the only change we found between the original and modified sub-regions. So, we compared observed in situ soil moisture in the Southeast sub-region using the original boundaries and the modified boundaries to evaluate whether changing the spatial extent of the sub-region had a significant impact on the observed soil moisture measurements. We plotted the area averaged monthly in situ soil moisture using the original boundaries and the modified boundaries in Fig.1.

The figure shows in both 0-10 cm and 0-100 cm soil layers, area-averaged soil moisture in the Southeast sub-region using the original boundaries is highly correlated with the soil moisture in modified Southeast sub-region. Therefore, we conclude that the modified sub-regions have relatively little impact on the area-averaged observed soil

moisture. In addition, both the modeled and observed soil moisture are calculated using the same boundaries. Therefore, the change in the regional boundaries does not affect the appropriateness of the model evaluation reported in this paper. For these reasons, we are confident that the modified regions used in this paper do not have a significant impact on the results.

Reference

Mei, R., and Wang, G.: Summer Land–Atmosphere Coupling Strength in the United States: Comparison among Observations, Reanalysis Data, and Numerical Models, Journal of Hydrometeorology, 13, 1010-1022, doi:10.1175/JHM-D-11-075.1, 2012.

2. Page 5, line 2. The authors mentioned that soil moisture data were collected from 8 different networks. Do the 8 networks use same way to measure soil moisture? If no, then is there any significant biases among networks?

Response:

Thanks for this great question. The eight networks use different methods to collect soil moisture data. In this study, the eight networks we used have been shown by Dirmeyer et al. (2016) to have relatively low random errors. The goal of this paper is to evaluate ESM simulated soil moisture using soil moisture observations. A detailed evaluation of the in situ networks is out of the scope of this study. However, we agree that differences between these networks may affect the results. Therefore, we have reported this issue in the limitations section of our paper to highlight potential future work.

Reference

Dirmeyer, P. A., Wu, J., Norton, H. E., Dorigo, W. A., Quiring, S. M., Ford, T. W., Santanello, J. A., Bosilovich, M. G., Ek, M. B., Koster, R. D., Balsamo, G., and Lawrence, D. M.: Confronting Weather and Climate Models with Observational Data from Soil Moisture Networks over the United States, Journal of Hydrometeorology, 17, 1049-1067, 10.1175/JHM-D-15-0196.1, 2016.

3. Comparison between point measurements and gridded value is a big challenge, especially in a big grid box. Can simple spatial average method solve the issue?

Response:

Thank you for the question. Due to the complex spatial variability of soil moisture, a simple spatial average is not the ideal approach to upscaling soil moisture. It may result in the loss of some spatial information. We realize using more advanced aggregation methods may improve the accuracy of this analysis. However, since our evaluation focused on a coarse temporal scale (monthly scale), the influence of the spatial aggregation method is less important. Spatial averaging is commonly used to compare station data to modeled data. For example, Xia et al. (2015) used state-wide averaged soil moisture from stations in Alabama, Colorado, and Oklahoma to validate NLDAS-2 model simulations.

Reference

Xia, Y., Ek, M. B., Wu, Y., Ford, T., and Quiring, S. M.: Comparison of NLDAS-2 Simulated and NASMD Observed Daily Soil Moisture. Part I: Comparison and Analysis, Journal of Hydrometeorology, 10.1175/JHM-D-14-0096.1, 2015.

4. Page 6, line 14. Add a space between "<" and "0.25".

Response:

This change has been made.

5. Section 3.1, page 8. The content in this page is about the evaluation of individual models. Generate another section to present these results.

Response:

This is a good suggestion. We created Section 3.2 to discuss the evaluation of individual models.

6. Page 27, Figure 9b. It is better to change 1 m to 100 cm at the top of the figure, because it is important to keep expressions consistent throughout the paper.

Response:

Thanks for the comment. "1 m" in Figure 9b has been changed to "100 cm".

[Figure]

[Figure]

**Fig. 1.** Spatial averaged monthly soil moisture in the original Southeast (blue) and in the modified Southeast (red). Upper (lower) figure shows soil moisture in 0-10 cm (0-1 m) soil layer.

---

## Author Comment (AC2) · 14 Feb 2017

RESPONSE TO THE REVIEWER #2'S COMMENTS

We appreciate the reviewer's positive comments, and agree with most of the suggestions. The comments have improved the manuscript. We have answered the questions and revised the manuscript carefully. Our responses are provided below.

General comments:

1. Both, the spatial downscaling of the CMIP5 output, and the gap filling of the in

situ soil moisture records, are - in my view - unnecessary data manipulations which could introduce errorneous signals to the raw data. Instead of the downscaling of the model data, I would recommend to upscale the observational data to the coarse spatial resolution of the CMIP5 models, especially because you only focus at the regional continental scale. Instead of the gap filling, I would compute the models' monthly averages using only the same days available in the observations. In any case I would expect some analyses indicating the impact of any data pre-processing you perform on the final results. Also, the satellite-based soil moisture data might not be available anywhere and anytime. The current manuscript does not mention how the authors deal with this.

Response:

Thanks for the comments. In this study, the spatial downscaling of the CMIP5 output (bilinear interpolation) is only applied for generating the CMIP5 ensemble. This is a common method to calculate model ensemble mean (Zhou et al., 2014; Chen and Frauenfeld., 2014; Joetzjer et al., 2013). It is necessary to use this approach because all of the models have a different spatial resolution. The comparison between in situ observations and model simulations already follows the reviewer's suggestion (upscale point measurements to model grid cell). We averaged all the in situ observations in each grid cell and compared this to model simulated value in that grid cell. We have clarified this in the methods section.

We selected 363 in situ sites over the CONUS based on the integrity of data. Most of the in situ sites (322 of 363) have complete daily soil moisture observations (no missing data). Rest of the sites (41 of 363) have missing data, but the days of missing data in each month are less than 5. The gap filling process was applied to months with < 5 days of missing data. The difference between the soil moisture before and after gap filling is minimal. We show the scatter plots between monthly soil moisture before and after gap filling in Fig.1. No significant differences are found. Therefore, we do not believe the gap filling procedure has a significant impact on the results of the paper.

The satellite data are not available/reliable when soils are frozen and/or ground is snow covered (page 5: line 27-28). Therefore, only warm season analyses are undertaken over the CONUS and northern U.S. sub-regions (page 5: line 28-29).

Reference

Zhou, B., Wen, Q. H., Xu, Y., Song, L., and Zhang, X.: Projected Changes in Temperature and Precipitation Extremes in China by the CMIP5 Multimodel Ensembles, Journal of Climate, 27, 6591–6611, doi: 10.1175/JCLI-D-13-00761.1., 2014.

Chen, L., and Frauenfeld, O. W.: A comprehensive evaluation of precipitation simulations over China based on CMIP5 multimodel ensemble projections, Journal of Geophysical Research: Atmospheres, 119, 5767-5786, 10.1002/2013JD021190, 2014.

Joetzjer, E., Douville, H., Delire, C., and Ciais, P: Present-day and future Amazonian precipitation in global climate models: CMIP5 versus CMIP3, Clim Dyn: 41: 2921. doi:10.1007/s00382-012-1644-1, 2013.

2. The comparison between the absolute soil moisture in the ECV data and the CMIP5 models is maybe not appropriate. I think the absolute soil moisture amount in the ECV data has been scaled using data from land surface model simulations, while only the temporal variations are a truly observed feature. This would mean that when comparing the spatial patterns of the absolute soil moisture contents you actually compare model against model.

Response:

Thanks for the comment. ECV soil moisture is a product based on remote sensing observations that are rescaled by the Noah Land Surface Model from Global Land Date Assimilation System (GLDAS). A cumulative distribution function (CDF) matching technique is employed so that the temporal pattern reflects what is observed by the satellite. In this study, except for Figure 5 (in the manuscript), the analyses are based on temporal patterns in soil moisture. Hence, the comparison between the VWC in

the ECV and the CMIP5 models reflects the performance of CMIP5 models relative to satellite observations. We agree the reviewer's comment that while comparing the spatial patterns of VWC (Figure 5, in the manuscript), the results are affected by the Noah model. Therefore, we have added some descriptions in the results section to clarify our results. Because GLDAS uses different forcing data and parameters from CMIP5 experiments, we think the ECV soil moisture is still an independent soil moisture data source that can be compared to CMIP5 and in situ soil moisture. Therefore, the results shown in Figure 5 (in the manuscript) are instructive and they are supported by the comparison in Figure 4 (in the manuscript).

3. I think the results of this study can be very useful to guide model development, as well as the development of large-scale observational soil moisture products. While I recognize that this is not the main goal of this study, I would like to see some more explanations why poor model performance or differences across in situ and satellite based soil moisture are seen at several of the performed comparisons. This could then lead into explicit advice for the developers of the models and the soil moisture products.

Response:

We appreciate the reviewer's positive comments on the value of this study. There are a lot of factors that influence the accuracy of soil moisture in ESMs, such as forcing data, coupling algorithm, structure and parameters of land surface scheme, representation of physical processes, spatial resolution, etc. Given the scope of this paper, we can only answer this question by summarizing some of the similarities and differences between the "better" and "worse" models. The land surface models used in the ESMs play a critical role in simulating soil moisture. CESM1, CCSM4 and GFDL-ESM2M (which all performed better based on Taylor's Skill score) divide 0-1 m soil column into 7, 7 and 10 layers, respectively. These models provide more detailed soil moisture simulations than CanESM2 and HadGEM2-ES (2 layers in 0-1 m soil layer; both performed poorly based on Taylor's Skill Score). Additionally, the spatial resolutions of CMIP5 models are also different. Coarser resolutions may also lead to lower skill because they cannot

capture for the spatial variability of soil moisture. Relative to CESM1 and CCSM4 (192*288), CanESM2 has much coarser spatial resolution (64*128) and much poorer performance. This information has been added to Section 3.1 of the paper. A detailed examination of the strengths and weaknesses of each model would require a process-level study. This is beyond the scope of this paper.

4. The results section can be significantly shortened. Description of results displayed in figures does not need to be so comprehensive.

Response:

Thanks for the suggestion. We believe that the length of the results section in this paper is reasonable given that it is similar to other published HESS articles. However, based on the reviewer's suggestion, we simplified some descriptions of our figures and kept only the information that is necessary.

5. While the manuscript is clearly structured and overall easy to read, there are many small language errors (such as missing articles). I recommend that the authors take special care of these when revising the manuscript.

Response:

Thanks for the comment. We have examined the paper again carefully and fixed all the language issues related to grammar and citations.

Specific comments:

Title:

6. ... over the contiguous United States...

Response:

We have revised the title.

Abstract:

7. line 15: maybe replace 'magnitude' with 'amount' through the manuscript

Response:

We have replaced 'magnitude' with 'amount'.

8. line 16: 'variations in model performance' could be spatial, temporal, or across models (which is what you mean, I guess). Please clarify.

Response:

We have clarified the use of the term 'variations' in the abstract.

9. line 16: 'especially in the near-surface', please replace with 'at' or 'for the near surface'

Response:

We have replaced 'in the near-surface' with 'at the near-surface'.

10. line 17: deeper soil layers

Response:

A change has been made.

page 2: line 5:

11. please explain 'ground fluxes'

Response:

Thanks for the comment. Ground flux is the downward heat flux into the subsurface medium.

12. line 8: remote sensing observations

Response:

Revision has been made.

13. line 21: change 'predict ... earlier in ...' to 'better predict'

Response:

A change has been made.

14. line 23: 'from satellites remote sensing', improve phrasing

Response:

Phrase has been changed to 'from satellite remote sensing'.

15. line 30: Why is it a problem that ASCAT soil moisture is influenced by precipitation and evaporation? Soil moisture is by definition influenced by these quantities.

Response:

Thanks for the question. We have changed 'soil moisture' to 'spatial variance of soil moisture'. Rötzer et al. (2015) found the spatial variance of satellite soil moisture is highly dependent on the retrieval methods of the respective products. They stated "retrieval method causes higher influence of temporal variant factors (e.g. precipitation, evaporation) on the ASCAT product, while SMOS and ERA products are stronger determined by temporal invariant factors (e.g. topography, soil characteristics)".

Reference

Rötzer, K., Montzka, C., and Vereecken, H.: Spatio-temporal variability of global soil moisture products, Journal of Hydrology, 522, 187-202, http://dx.doi.org/10.1016/j.jhydrol.2014.12.038, 2015

16. line 32: abbreviation AMSR-E not introduced

Response:

Thanks for the comment. 'AMER-E' has been changed to 'the Advanced Microwave

[Figure]

Scanning Radiometer for EOS (AMSR-E)'.

page 3:

17. line 7: 'offline land surface models', please improve phrasing

Response:

Thanks for the comment. 'offline land surface models' has been changed to 'land surface models'.

18. line 8: at a variety of depths

Response:

Fixed.

19. line 9: biases

Response:

Fixed.

20. line 11: please correct citation style

Response:

Thanks for the comment. Citation style has been corrected.

21. line 22: difficulties to accurately simulate

Response:

Fixed.

22. line 28: abbreviation CONUS not introduced

Response:

Thanks for the comment. 'CONUS' has been changed to 'the contiguous United States

(CONUS)'.

23. line 31: ... followed by the presentation of the results and a discussion in section 3. Limitations and conclusions of the study are ...

Response:

Thanks for pointing out the errors. Changes has been made.

page 4: 24. section 2.1: How do the adjustments of the region perimeters influence the results of the study?

Response:

Thanks for this question. We have generally used the same sub-regions as in previous studies, but four sub-regions (Northern Great Plains, Southern Great Plains, Midwest and Southeast) were modified so that we could include more in situ soil moisture measurements. The original boundaries of each sub-region were based on the land cover types and were applied in soil moisture related research (Mei and Wang, 2012). We found the land cover types in the modified sub-regions do not change greatly. In Northern and Southern Great Plains, the main effect of the modification is including more soil moisture sites in eastern Oklahoma where the dominate land cover type is Savanna. Savanna is also the main land cover type in central Oklahoma. In the Midwest, the modified region expands northward. Most of the sites (5 of 6) that were added are located in cropland region. This is consistent with the dominant land cover, since cropland covers more than 90% area of the Mideast. In the original Southeast, evergreen forest is the main land cover type. The land cover types in the modified Southeast is mixed by evergreen forest, deciduous forest and mixed forest. This is the only change we found between the original and modified sub-regions. So, we compared observed in situ soil moisture in the Southeast sub-region using the original boundaries and the modified boundaries to evaluate whether changing the spatial extent of the sub-region had a significant impact on the observed soil moisture measurements. We plotted

the area averaged monthly in situ soil moisture using the original boundaries and the modified boundaries in Fig.2.

The figure shows in both 0-10 cm and 0-100 cm soil layers, area-averaged soil moisture in the Southeast sub-region using the original boundaries is highly correlated with the soil moisture in modified Southeast sub-region. Therefore, we conclude that the modified sub-regions have relatively little impact on the area-averaged observed soil moisture. In addition, both the modeled and observed soil moisture are calculated using the same boundaries. Therefore, the change in the regional boundaries does not affect the appropriateness of the model evaluation reported in this paper. For these reasons, we are confident that the modified regions used in this paper do not have a significant impact on the results.

Reference

Mei, R., and Wang, G.: Summer Land–Atmosphere Coupling Strength in the United States: Comparison among Observations, Reanalysis Data, and Numerical Models, Journal of Hydrometeorology, 13, 1010-1022, doi:10.1175/JHM-D-11-075.1, 2012.

25. section 2.2: Why was this particular emission scenario used here?

Response:

Thanks for this question. We pick up RCP4.5 based on two reasons (page 4, line19-23). First, in the latest IPCC AR5 report (Bindoff et al., 2013), RCP4.5 scenario is used to extend the CMIP5 historical experiment. Specifically, Figure 10.1 to Figure 10.3 and Table 10.SM.2 are created based on these methods. Second, we compared the CRU precipitation with RCP2.6, RCP4.5 and RCP8.5 precipitation, as shown in Tab.1 (see supplement), which shows that RCP4.5 has relatively small bias and the most similar variance.

Reference

Bindoff, N. L., Stott, P. A., AchutaRao, K. M., Allen, M. R., Gillett, N., Gutzler, D.,

Hansingo, K., Hegerl, G., Hu, Y., Jain, S., Mokhov, I. I., Overland, J., Perlwitz, J., Sebbari, R., and Zhang, X.: Detection and Attribution of Climate Change: from Global to Regional. In: Climate Change 2013: The Physical Science Basis. Contribution of Working Group I to the Fifth Assessment Report of the Intergovernmental Panel on Climate Change, Cambridge, United Kingdom and New York, NY, USA., 2013

26. line 3: please refer to 'eight sub-regions' instead of 'regions' through the manuscript (as also CONUS is a region)

Response:

Thanks for the comment. Changes have been made.

27. lines 7-10: a table would be suitable to present these information;

Response:

Thanks for the comment. The longitude/latitude information provided here is supplementary information to the Figure 1. We do not think it is necessary to use a table here.

28. line 12: provide soil moisture data

Response:

We do not understand this comment.

29. line 15: 'with the greater number of in situ observations', please improve phrasing

Response:

Thanks for the comment. 'this is the time period with the greater number of in situ observations' has been changed to 'more in situ sites are available in this 10-year time period'.

30. lines 16-18: please improve phrasing

Response:

Thanks for the comment. The sentence has been revised.

31. line 25: using a bilinear interpolation method Response: Fixed.

32. line 30: Daily in situ soil moisture data from ...

Response:

Fixed.

page 5: 33. section 2.3: Do your results depend on the amount of stations in each sub-region?

Response:

Thanks for the question. Theoretically, the answer is yes. However, this dependence does not affect the results of our study. The comparison in each sub-region is very straightforward. We only use the grid cells with in situ sites in them. This method is a common way to show the moisture conditions over a region. For example, the National Drought Mitigation Center uses the same approached to calculate state-wide average soil moisture conditions. In addition, previous studies have also used this approach to represent state-wide soil wetness. For example, Xia et al, (2015) uses the state-averaged soil moisture in Alabama, Colorado, and Oklahoma to validate NLDAS-2 model simulations. Hence the comparison is not affected by the spatial distribution of in situ sites. The only issue related to the amount of stations in each sub-region is how well the in situ observation represents the soil moisture condition in each sub-region. We have reported this in the limitation section (line 26-28, page 10).

Reference

Xia, Y., Ek, M. B., Wu, Y., Ford, T., and Quiring, S. M.: Comparison of NLDAS-2 Simulated and NASMD Observed Daily Soil Moisture. Part I: Comparison and Analysis, Journal of Hydrometeorology, 10.1175/JHM-D-14-0096.1, 2015.

34. section 2.4: What version of the ESA CCI soil moisture is employed?

Response:

Thanks for the question. We used the CCI soil moisture v02.2. This is the latest version we can find on the CCI website.

35. Maybe an upgrade (if possible) would improve the coherence between the in situ and satellite derived soil moisture results?

Response:

Thanks for the comment. This product is the latest one available on the CCI website. The goal of this study is to evaluate soil moisture from CMIP5 models using two independent data sources. We are not aiming to compare in situ and satellite observations. Both in situ or satellite observations have strengths and weaknesses. This study does not address which is more accurate. The differences between in situ and satellite observations are interesting and that future research should undertake a detailed comparison.

36. line 1-2: These stations belong to eight ...

Response:

Fixed.

37. line 2: Quality-controlled daily soil moisture data have ...

Response:

Fixed.

38. line 4: gap filling of missing data: Beside the main comment above, how does it work?

Response:

Thanks for the comment. The approach we used in this study is called Daily Average Replacement (DAR) method. The DAR method fills missing values using observations from before and after the missing day. In this study, we use 5 days before and after the missing day. This approach was developed by Dumedah and Coulibaly (2011) for infilling soil moisture data in southern Ontario, Canada. Based on Ford et al. (2014), DAR outperforms other methods for replacing missing soil moisture data (e.g., coefficient of correlation weighting, inverse distance weighting, ordinary kriging, and spatial regression).

Reference

Dumedah G, and Coulibaly P.: Evaluation of statistical methods for infilling missing values in high-resolution soil moisture data, J. Hydrol. 400: 95–102, 10.1016/j.jhydrol.2011.01.028, 2011.

Ford, T. W., and Quiring, S. M.: Comparison and application of multiple methods for temporal interpolation of daily soil moisture, International Journal of Climatology, 34, 2604-2621, 10.1002/joc.3862, 2014.

39. line 15: 'provides an overview of soil moisture simulations in CMIP5 models'?

Response:

Thanks for the comment. Phrase has been changed to 'provides a general soil moisture conditions simulated by CMIP5 models over the CONUS'.

40. line 19: For a regional evaluation...

Response:

Fixed.

41. page 6: line 22: remove 'starting'

Response:

Fixed.

42. page 7: line 1: agreement in terms of what?

Response:

Thanks for the comment. 'agreement' has been changed to 'agreement on seasonal pattern of soil moisture'.

43. line 5: 'which is similar to the 0-10 cm soil layer', please improve phrasing, maybe use 'Similar results are found for the 0-10 cm soil layer.'

Response:

Thanks for the comment. Expression has been improved.

44. line 18: and of the negative biases

Response:

Fixed.

45. line 26: ECV shows more spatial heterogeneity

Response:

Fixed.

46. line 34: '(regions with a wet bias)'? remove?

Response:

Thanks for the comment. '(regions with a wet bias)' has been removed.

47. page 8: line 25: in the 0-10 cm soil layer

Response:

Fixed.

48. page 9: line 5: please rephrase

Response:

Thanks for the comment. The sentence has been changed to 'Only in the Northern Shrubland and Southern Shrubland regions, the MAE is lower when comparing to the in situ observations'.

49. page 10: line 7: the driest conditions

Response:

Fixed.

50. line 20: is more strongly correlated

Response:

Fixed.

51. page 11: line 14: comparatively dry 'substantial bias in the deeper soil layer', can you speculate why that is?

Response:

Thanks for the great question. Surface soil moisture is most strongly controlled by meteorological forcing (precipitation and evaporation). Soil moisture in the deeper soil is more strongly controlled by soil characteristics and soil physics. The model generalizes the deeper soil layer more than the surface layer. Soil texture associated parameters, such as porosity and hydraulic conductivity in model may be different than what exists at the measurement sites. This may cause larger bias in the deeper soil layer than in the top soil layer.

52. line 17: the observed spatial pattern

Response:

Fixed.

53. line 21: varies significantly across sub-regions

Response:

Fixed.

54. line 27: the CMIP5 ensemble

Response:

Fixed.

55. line 31: 'relatively consistent', not in the SS sub-region

Response:

Thanks for the comment. The 'relatively consistent' is used to describe the shapes of mean monthly ECV soil moisture and in situ soil moisture. In SS region, the in situ soil moisture has a larger seasonal variance than ECV soil moisture, both of the seasonal patterns show similar shapes of the ups and downs.

56. page 12: line 3: point out that

Response:

Fixed.

57. Figure 1, caption: and the boundaries of

Response:

Fixed.

58. Figure 2: Please use the same x and y-axes in all plots.

Response:

Thanks for the comment. Figure 2 (in the manuscript) has been re-plotted using same x and y-axes.

59. Figure 3, caption: CMIP5 ensemble mean (black line)

Response:

Fixed.

60. Figure 4: Maybe add white color in the middle of the color bar such that locations with good agreement do not show up?

Response:

Thanks for the comment. Figure 4e, f and 5e, f (in the manuscript) show the differences between model-simulated and observed soil moisture in each site or grid cell. We did not use white color in the figure is because we want to show wet/dry bias of model clearly. Adding white color in the color bar requires us to define a section that corresponds to "good agreement". However, it is not possible to find a uniform section for all the locations based on statistical test. Hence, we prefer to keep the color scheme.

61. Figure 9: Please label color bars. I find it interesting that the correlations for the SS sub-region are consistently low, whereas in Figure 8 the correlations for the model ensemble mean in that sub-region are high. Can you comment on that?

Response:

Thanks for the comment. Figure 9 (in the manuscript) shows the Taylor's Skill Score for each individual model. They not only reflect the correlation, but also describes the relative standard deviation, see Eq. 1. (in the manuscript) High correlation does not equal to high skill score. For example, in Figure 7(a) (in the manuscript), BCC model has low correlation coefficient ($\sim$0.45). Its corresponding Taylor's Skill Score is higher than 0.7 because its normalized standard deviation is close to 1. On the other hand, high correlation for the model ensemble does not mean high correlation

for the individual model. We found large variations across the models in Figure 3 (in the manuscript) even though the ensemble mean matches the observations very well. We have added the label in Figure 9 (in the manuscript) to clarify the content.

Please also note the supplement to this comment: http://www.hydrol-earth-syst-sci-discuss.net/hess-2016-477/hess-2016-477-AC2-supplement.pdf
* * *
[Figure]

[Figure]

**Fig. 1.** Scatter plots of soil moisture before and after gap filling.

[Figure]

**Fig. 2.** Spatial averaged monthly soil moisture in the original Southeast (blue) and in the modified Southeast (red). Upper (lower) figure shows soil moisture in 0-10 cm (0-1 m) soil layer.

**Supplement:**

Tab.1. Comparison of precipitation between CRU and different scenarios.

| Precipitation (mm) | CRU TS2.2 | RCP2.6 | RCP4.5 | RCP8.5 |
|---|---|---|---|---|
| Mean | 70.1 | *72.8* | *72.8* | 75.9 |
| Stand Deviation | 13.47 | 10.86 | *11.62* | 9.33 |